# PigSNIPE: Scalable Neuroimaging Processing Engine for Minipig MRI

Michal Brzus [1], Kevin Knoernschild [2,3], Jessica C. Sieren [2,3] and Hans J. Johnson [1,2,*]

[1] Department of Electrical and Computer Engineering, University of Iowa, Iowa City, IA 52242, USA
[2] Department of Biomedical Engineering, University of Iowa, Iowa City, IA 52242, USA
[3] Department of Radiology, University of Iowa, Iowa City, IA 52246, USA
[*] Correspondence: hans-johnson@uiowa.edu

**Abstract:** Translation of basic animal research to find effective methods of diagnosing and treating human neurological disorders requires parallel analysis infrastructures. Small animals such as mice provide exploratory animal disease models. However, many interventions developed using small animal models fail to translate to human use due to physical or biological differences. Recently, large-animal minipigs have emerged in neuroscience due to both their brain similarity and economic advantages. Medical image processing is a crucial part of research, as it allows researchers to monitor their experiments and understand disease development. By pairing four reinforcement learning models and five deep learning UNet segmentation models with existing algorithms, we developed PigSNIPE, a pipeline for the automated handling, processing, and analyzing of large-scale data sets of minipig MR images. PigSNIPE allows for image registration, AC-PC alignment, detection of 19 anatomical landmarks, skull stripping, brainmask and intracranial volume segmentation (DICE 0.98), tissue segmentation (DICE 0.82), and caudate-putamen brain segmentation (DICE 0.8) in under two minutes. To the best of our knowledge, this is the first automated pipeline tool aimed at large animal images, which can significantly reduce the time and resources needed for analyzing minipig neuroimages.

**Keywords:** minipig; brain; segmentation; landmarks; image processing; deep learning; pig





## 1. Introduction

Neurological disorders are common and predicted to affect one in six people [1]. Scientists constantly work to find more effective methods of diagnosing and treating these devastating conditions. Animal subjects are invaluable in neuroscience, as they allow researchers to induce diseases and their pathogenesis [2]. Small animals, most commonly mice, led to the development of many important animal models [3]. However, small animals lack the brain complexity and biological scale found in humans, resulting in many therapies failing to translate to human trials [4,5]. This issue led to an increased focus on developing large animal models. Animals such as sheep, pigs, and non-human primates resulted in more robust findings [6]. Recently, minipigs have emerged at the forefront of neuroscience, as they present multiple advantages over other large animals, such as a relatively large gyrencephalic brain that is similar to the human brain's anatomy, neurophysiological processes, and white-to-gray matter ratio (60:40) [7]. Furthermore, the minipig's use presents a considerably lower cost and fewer ethical issues [8] when compared with its alternatives.

Medical imaging, such as magnetic resonance (MR) imaging, provides researchers with non-invasive access to the brain. MRI produces higher-quality images compared with computed tomography without radiation risks, making it suitable for research. Medical image processing is a crucial part of research, as it allows researchers to monitor their experiments and understand disease development. Processes such as image registration, skull stripping, tissue segmentation, and landmark detection are necessary for many experiments.

In recent years, deep learning has overtaken the field by outperforming virtually all previous algorithms and allowing for solving entirely new problems. Nowadays, most new AI developments, such as general adversarial networks and transformers, also succeed in medical imaging [9,10]. However, many algorithms are created and optimized for MR analysis of human data and are not directly applicable or sufficiently sensitive to measure large-animal data. For example, due to differences in the size and shape of the head, highly successful tools such as SynthStrip [11] fail on minipig images. This forces researchers into laborious, expensive, and error-prone manual processing. Therefore, there is an urgent need for accurate and automated tools to analyze minipig data.

In this work, we propose PigSNIPE, a pipeline for the automated processing of minipig MR images similar to software available for humans [12]. This is an extension of our previous work [13] and allows for image registration, AC-PC alignment, brain mask segmentation, skull stripping, tissue segmentation, caudate-putamen brain segmentation, and landmark detection in under two minutes. To the best of our knowledge, this is the first tool aimed at animal images which will dramatically reduce the time and resources needed to analyze minipig data and accelerate the scientific discovery process. Our tool is open source and can be found at https://github.com/BRAINSia/PigSNIPE (accessed on 6 February 2023).

## 2. Data

In this study, we used two large-animal minipig datasets. The first (Germany) dataset consists of 106 scanning sessions from 33 Libechov minipigs [14] collected using a 3T Philips Achieva scanner. Each scanning session contained single T1 and T2 weighted scans. The T1w images were acquired at the acquisition resolution of $0.5 \times 0.514 \times 0.514$ mm$^3$ and a spatial size of $450 \times 576 \times 576$ voxels. The T2w images have a $0.1875 \times 0.1875 \times 2.2$ mm$^3$ resolution and a spatial size of $960 \times 960 \times 30$ voxels.

The second (Iowa) dataset was collected for a study investigating CLN2 gene mutation at the University of Iowa [15]. The Iowa dataset consisted of 38 scanning sessions from 23 Yucatan minipigs using a 3.0T GE SIGNA Premier scanner. Many scanning sessions contain more than one T1w and T2w image, resulting in a total of 178 T1w and 134 T2w images. Both the T1w and T2w images were acquired at a resolution of $0.7 \times 0.625 \times 0.625$ mm$^3$ and a spatial size of $104 \times 256 \times 256$ voxels.

Based on our data, we created two training datasets. The first dataset was used for the brainmask training models. As both low-resolution and high-resolution brainmask models worked on either the T1w or T2w images, we took advantage of all available data, as shown in Table 1. The second dataset was used to train the intracranial volume, white and gray matter and CSF, and caudate-putamen segmentation models, which require as input the registered T1w and T2w images. We computed a Cartesian product of all T1w and T2w images for each scanning session to maximize our training dataset and obtain all possible pairs. The range of all possible pairs per scanning session was between 1 and 28. Therefore, to avoid the overrepresentation of some subjects, we limited the number of T1w-T2w pairs to four. The resulting training split is in Table 2. For each dataset, we split the data by subject into training, validation, and test sets (80%, 10%, and 10%, respectively) on the subject basis to ensure data from the same animal appeared only in one subset.

**Table 1.** Training dataset low-resolution and high-resolution brainmask models.

| Number of | Germany Dataset | | Iowa Dataset | | Total | |
|---|---|---|---|---|---|---|
| | Images | Subjects | Images | Subjects | Images | Subjects |
| Training | 164 | 26 | 243 | 18 | 407 | 44 |
| Validation | 30 | 4 | 39 | 3 | 69 | 7 |
| Test | 18 | 3 | 30 | 2 | 48 | 5 |

**Table 2.** Training dataset for segmentation models.

| Number of | Germany Dataset | | Iowa Dataset | | Total | |
|---|---|---|---|---|---|---|
| | Image Pairs | Subjects | Image Pairs | Subjects | Image Pairs | Subjects |
| Training | 78 | 26 | 113 | 18 | 191 | 44 |
| Validation | 15 | 4 | 19 | 3 | 34 | 7 |
| Test | 8 | 3 | 14 | 2 | 22 | 5 |

In addition to the original T1w and T2w scan data, we generated inter-cranial volume (ICV) mask, caudate-putamen (CP), and gray and white matter and cerebrospinal fluid (GWC) segmentations. The ICV masks served as the ground truth for low-resolution, high-resolution, and ICV models. The CP segmentations were manually traced and used to train the CP segmentation model. The GWC masks were generated using Atropos software [16] and used for training the GWC model. The physical landmark data consisted of 19 landmarks used for training the landmark detection models. Detailed information on the data and implementation of landmark detection can be found in our previous work [13].

## 3. Materials and Methods

### 3.1. Pipeline Overview

Figure 1 displays the simplified view of the PigSNIPE pipeline. The pipeline consists of five main parts, and the flow of the different data types is indicated by the line styles described in the legend. The pipeline's inputs are T1w and T2w images (middle of the figure) in their original physical spaces. The starting step (Brainmask) is to compute the image brainmasks as other processes require them. First, we compute a low-resolution brainmask to crop the original image around the brain. This step then allows for computing a high-resolution brainmask. Then, the original T1w and T2w images and their corresponding high-resolution brainmasks are used in the registration process (Registration). The co-registered T1w and T2w images are used in the Segmentation part to compute the intracranial volume mask (ICV), gray and white matter and cerebrospinal fluid mask (GWC), and caudate-putamen segmentation (Seg). On the other side of the pipeline, the Landmark Detection step uses the T1w image with its high-resolution brainmask. This process produces the ACPC transform and ACPC-aligned T1w image, and it computes 19 physical landmarks saved in both the original T1w and ACPC spaces. The last part of the pipeline is ACPC alignment. At this point, the computed ICV, GWC, Seg, and T2w registered images are in the original T1w space. Therefore, we can reuse the ACPC transform computed during the Landmark Detection process to resample the data into the ACPC space. The figure displays the full configuration of the pipeline. However, the user can choose the configuration, such as not running the landmark detection and AC-PC alignment, which greatly reduces the runtime. The user can also decide to skull-strip the data (not shown in the figure).

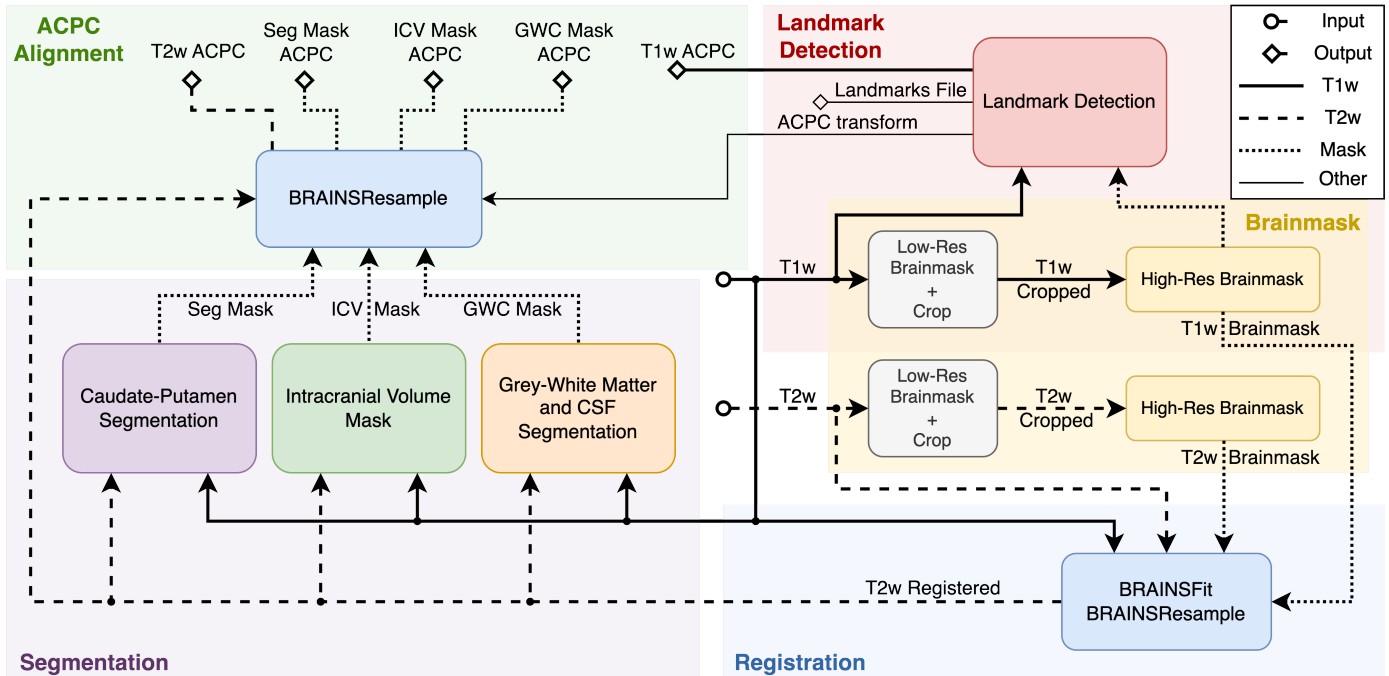

**Figure 1.** Simplified view of the pipeline's dataflow. Input T1w and T2w images undergo brainmask generation in two steps to allow for robust image registration. Segmentation models use registered images to generate an intracranial volume mask, gray and white matter and cerebrospinal fluid tissue segmentation, and caudate and putamen masks. T1w image is also used to detect 19 anatomical landmarks and AC-PC transform, allowing for standardized alignment of all data.

### 3.2. Deep Learning Segmentation Model

All brainmasks and segmentations are based on the same model architecture: the 3D ResUNet architecture [17] (see the citation for the architecture details). Each model uses the Adam optimizer, a learning rate of 0.001, and the DiceCE loss, which combines the Dice and cross-entropy losses. The implementation leveraged the existing MONAI [18], PyTorch [19], and Lightning [20] libraries.

All models used 5 layers with 16, 32, 64, 128, and 256 channels per layer (with the low-resolution model being an exception, having 128 channels in the last layer) and 3 residual connections. The only variations between the models were in image voxel size. In the low-resolution model, we resampled a $64 \times 64 \times 64$ voxel region with a 3 mm isotropic spacing from the center of the image. This dramatically decreased the image size by over 95%, allowing for easy model training. All other models used an isotropic spacing of 0.5 mm. The high-resolution model used a $224 \times 224 \times 160$ region, the ICV and GWC models used a $128 \times 224 \times 160$ region, and the Seg model used a $98 \times 126 \times 96$ voxel region. Those regions were resampled around the centroid of the low-resolution brainmask for the high-resolution model, high-resolution brainmask for the ICV model, and ICV mask for the GWC and Seg models. The centroids were computed using ITK's ImageMomentsCalculator [21]. Additionally, to make the high-resolution brainmask model robust to errors in the centroid position resulting from the inaccuracy of the low-resolution model, we added a random vector to the computed centroid position during training.

For preprocessing, all models used intensity value scaling from $-1.0$ to $1.0$ with truncation on the 1% and 99% percentiles and implemented standard data augmentation techniques: random rotation, zoom, noise, and flip (except for the Seg model). Additionally, for the ICV, GWC, and Seg models, we passed both T1w and T2w images on two channels.

For post-processing during inference, we applied the FillHoles transform, which would fill all holes that could randomly occur in the predicted masks for all models. Addi-

tionally, all models except for the GWC model used KeepLargestConnectedComponent, which removed the noise outside the predicted mask.

### 3.3. Landmark Detection

A fundamental step in neuroimage analysis is anatomical landmark detection. Landmarks have many uses in medical imaging, including morphometric analysis [22], image-guided surgery [23], or image registration [24]. In addition, many analysis tools require landmarks to co-register different imaging sets.

The landmark detection is a two-stage process involving four deep reinforcement learning models [25]. All models detect multiple landmarks simultaneously through the multi-agent, hard parameter-sharing [26,27] deep Q network [28]. The models are optimized by Huber loss [29] and an Adam optimizer [30] and controlled by an $\epsilon$-greedy exploration [31] policy.

The first step was to crop the T1w image and compute the brain mask's center of gravity, which is used for reinforcement learning initialization. In the first two models, we computed three landmarks, allowing AC-PC alignment of the T1w image. After that, the image in the standard AC-PC space allowed for accurate computation of the remaining 16 landmarks.

Detailed information about the models' architecture, training, and results can be found in our previous work [13].

### 3.4. Image Registration and AC-PC Alignment

Despite images from the same scanning session being taken minutes apart, many sources for possible image misalignment exist. For example, the animal might move during the scanning session, resulting in poor alignment of the images. As the ICV, GWC, and Seg models use both T1w and T2w images simultaneously, it is crucial to ensure proper image registration.

For all registrations, we used the BRAINSFit and BRAINSResample algorithms from the BRAINSTools package [32]. The algorithms use high-resolution masks to ensure that the registration samples data points from the region of interest. By default, the pipeline uses Rigid registration with ResampleInPlace. This allows for image registration without interpolation errors. However, the user has the option to use the Rigid+Affine transform, which allows for nonlinear transformation.

Additionally, the pipeline can transform all data into a standardized anterior commissure-posterior commissure ( AC-PC) aligned space. This process combines the ACPC transform generated during the landmark detection process and the T2w-to-T1w transform in native space to achieve a T2w image in ACPC space. As all computed segmentations were in the T1w image, we could apply the ACPC transform to generate ACPC-aligned masks.

## 4. Results

### 4.1. Deep Learning Model Accuracy

We evaluated the model performance using DICE [33] and the balanced Hausdorff distance [34]. DICE is a similarity coefficient that measures the overlap over the union of two binary masks, and the Hausdorff distance measures the average distance error between the ground truth and the predicted masks. To compute the metrics, we used the Evaluate Segmentation tool [35] and TorchMetrics package [36].

#### 4.1.1. Brainmask

Figure 2 shows a 2D sagittal slice of the T1w and T2w images and their corresponding low- and high-resolution brainmasks. We can see that the low-resolution mask had very coarse boundaries and mistakes, such as missed parts of the cerebellum, indicated by the red arrow. Although the accuracy of the low-resolution mask was much worse than that of the high-resolution mask, it robustly found the position of the brain. This allows for accurate image cropping and the generation of a highly accurate high-resolution mask.

| Modality | Image | Low-Res Mask | High-Res Mask |
|----------|-------|--------------|---------------|

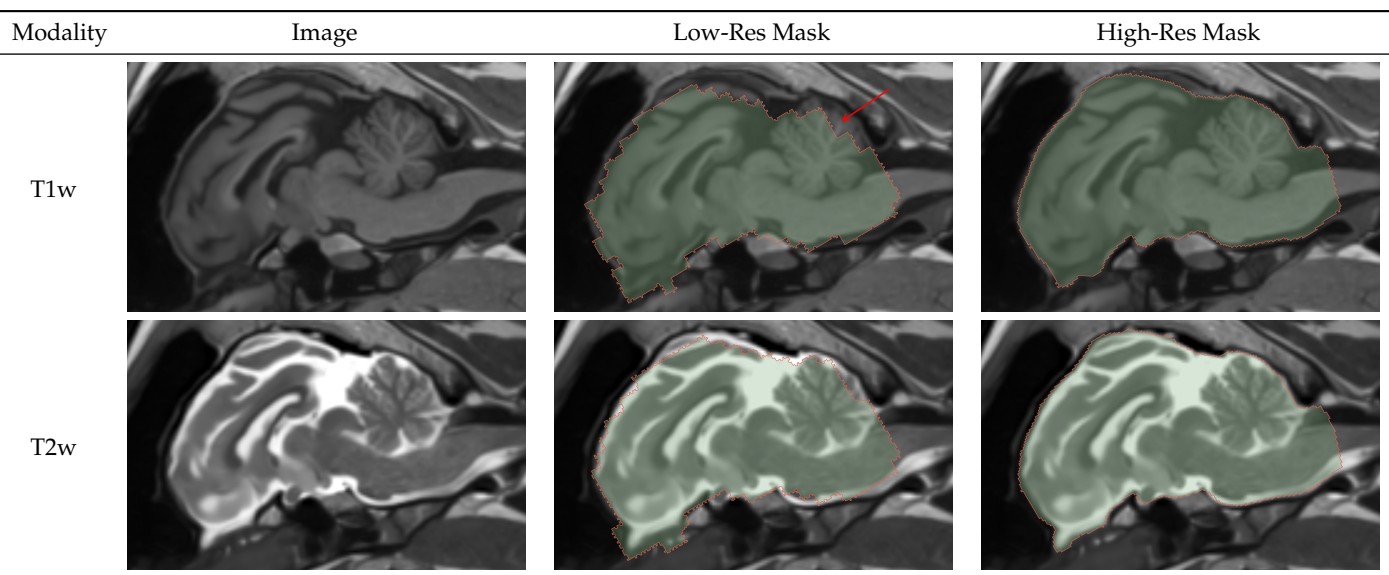

**Figure 2.** Result visualization for low and high-resolution brainmask models on T1 and T2 weighted images. The low-resolution model robustly detected the brain from a large volume, allowing for image cropping and generating an accurate high-resolution brainmask.

Table 3 shows the DICE coefficient and balanced average Hausdorff distance (bAVD) results on the test set. We can see that SynthStrip—a state-of-the-art algorithm for human data—failed with the minipig data, while our models performed well with both the Germany and Iowa data. The low-resolution model achieved a DICE score of 0.88 with a standard deviation and bAVD of 0.25 voxels, which proves the robustness of the model's ability to find the brain. The high-resolution model was highly accurate, with an average distance error of 0.04 voxels and DICE of 0.97. Both models were consistent, with a standard deviation of 0.02 for DICE and under 0.2 voxels for the bAVD.

**Table 3.** Accuracy evaluation for SnthStrip, low-resolution, and high-resolution brainmask models. SynthStrip failed to perform with minipig data. The low-resolution model performed well in terms of both DICE and bAVD metrics, proving its robust brain position estimation using severely down-sampled data. The high-resolution model achieved very high accuracy for both metrics, supporting two-step brainmask generation.

| | SynthStrip | | Low-Resolution | | High-Resolution | |
|---------|------|--------|------|------|------|------|
| **Dataset** | **DICE** | **bAVD** | **DICE** | **bAVD** | **DICE** | **bAVD** |
| Iowa | 0.32 | 74.58 | 0.89 | 0.21 | 0.97 | 0.03 |
| Germany | 0.19 | 152.52 | 0.88 | 0.32 | 0.97 | 0.06 |
| Total | 0.23 | 128.33 | 0.88 | 0.25 | 0.97 | 0.04 |

4.1.2. Intracranial Volume Mask

Figure 3 shows the input T1w and T2w images, the ground truth, and the predicted ICV mask. As can be seen in Table 4, the ICV model was more accurate and consistent than the high-resolution model at a DICE of 0.98, with a standard deviation of 0.003 and bAVD of 0.03 voxels and a standard deviation of 0.004 voxels. Furthermore, those numbers were negatively impacted by the posterior mask cutoff differences, indicated by the red arrow.

| T1w | T2w | Ground Truth | Our Model |

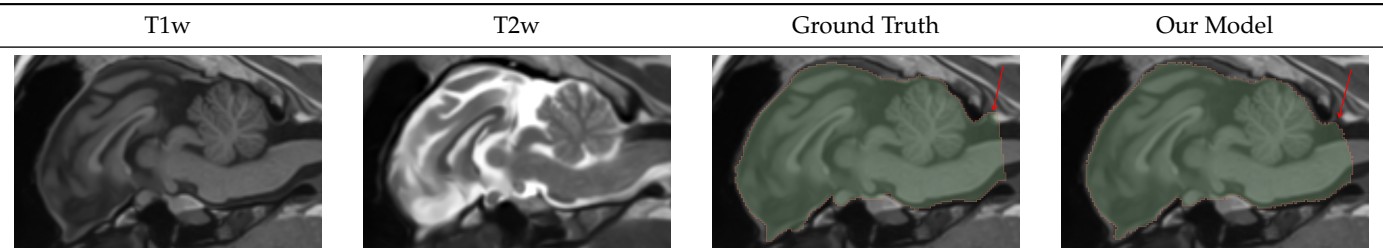

**Figure 3.** Intracranial volume mask visualization. Using both T1 and T2 weighted images simultaneously, our model generated exceptionally high-quality masks that only differed from the ground truth in the posterior cutoff point, indicated by the red arrow.

**Table 4.** DICE and balanced average Hausdorff distance results for intracranial volume model. DICE score indicates extremely high similarity to the ground truth images in both the Iowa and Germany datasets. Likewise, the low-balance average Hausdorff distance proves the accuracy of our model.

| Dataset | DICE | bAVD |
|---------|------|------|
| Iowa | 0.97 | 0.03 |
| Germany | 0.98 | 0.03 |
| Total | 0.98 | 0.03 |

### 4.1.3. Caudate-Putamen Segmentation Mask

Figure 4 shows a portion of an axial slice of the input T1w image, the ground truth manual segmentation, our Seg model prediction, and the difference between them. In the T1w image, we can see that the caudate and putamen regions are very hard to distinguish, requiring an expert to label them correctly. We can see that although the boundaries of the predicted regions varied from the ground truth, our model prediction was very close to the ground truth. In the qualitative analysis, we observed that the Seg model tended to over-segment the regions. Table 5 shows the metrics for the Seg model performance on the test dataset. We can see that all regions had a similar DICE score of 0.8 and a bAVD of around 0.28 voxels. The model was consistent on the Iowa dataset, with a standard deviation under 0.05 for the DICE for all classes and a standard deviation of 0.06 voxels for the bAVD. The model consistency on the German dataset was slightly worse, as indicated by the standard deviation of 0.15 voxels for the bAVD and 0.07 for the DICE for the left putamen. We believe this was due to the worse image quality of the German T2w images.

| T1w | Ground Truth | Our Model | Difference |

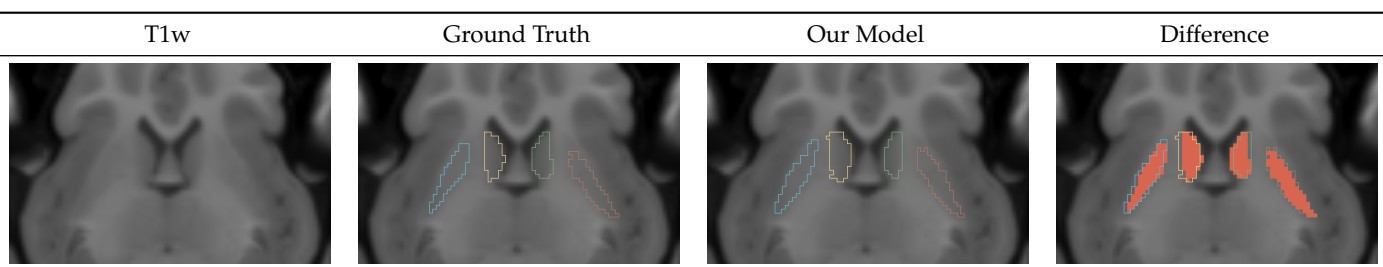

**Figure 4.** Caudate-putamen segmentation mask visualization. The difference image shows the ground truth mask in red with the color outline of our model prediction. The mask generated by our model closely resembles the manually traced ground truth label.

**Table 5.** DICE and balanced average Hausdorff distance results for caudate-putamen segmentation model. The model performed consistently, with a DICE of around 0.8 and bAVD of 0.28 voxels.

| Dataset | Left Caudate | Right Caudate | Left Putamen | Right Putamen | Global bAVD |
|---------|--------------|---------------|--------------|---------------|-------------|
| Germany | 0.82 | 0.79 | 0.79 | 0.81 | 0.29 |
| Iowa | 0.80 | 0.83 | 0.78 | 0.80 | 0.27 |
| Total | 0.81 | 0.82 | 0.78 | 0.80 | 0.28 |

### 4.1.4. Gray-White-CSF Segmentation Mask

Figure 5 shows an axial slice of an example T1-weighted image, Atropos prediction, and our model prediction. Atropos is a tool aimed at human data, and the results produced for the minipig data were suboptimal, making the comparison harder. We can see that the produced masks were very similar. We observe that Atropos tends to oversegment the white matter, especially in the cerebellum region.

| T1w | Atropos | Our Model |
|-----|---------|-----------|

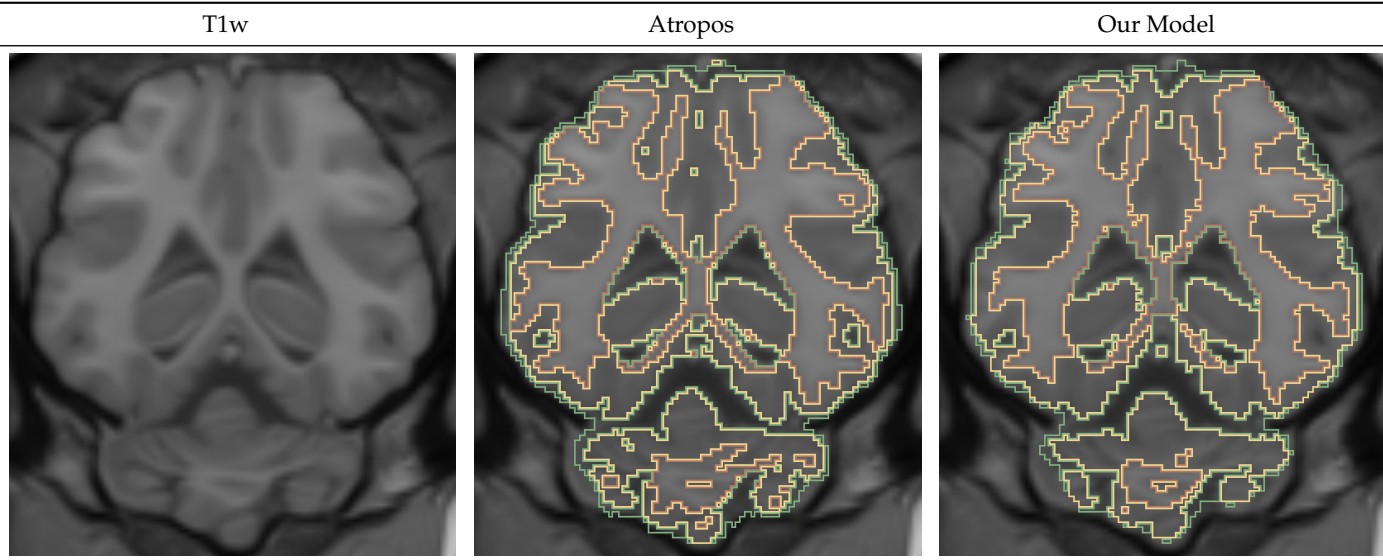

**Figure 5.** Gray and white matter and cerebrospinal fluid segmentation mask visualization. Our model produced masks similar to the Atropos tool.

Table 6 shows the DICE and bAVD results for the GWC model performance on the test dataset compared with the Atropos results. We can see that the CSF performance was consistently the lowest and that the white matter had the highest DICE value. We observed a low distance error of bAVD 0.05 voxels for all masks, with a standard deviation of 0.01 voxels. Additionally, the results were hurt by the posterior cutoff point in the same way as in the ICV model, as both Atropos and our GWC model used the ICV mask as input.

**Table 6.** Gray and white matter and cerebrospinal fluid segmentation similarity to Atropos. White matter was consistently the most similar between two models, with a DICE score of 0.89. The distance erros were consistant at 0.05 voxels across the test dataset.

| Dataset | Gray Matter | White Matter | CSF | Global bAVD |
|---------|-------------|--------------|-----|-------------|
| Germany | 0.79 | 0.89 | 0.71 | 0.06 |
| Iowa | 0.81 | 0.89 | 0.78 | 0.05 |
| Total | 0.80 | 0.89 | 0.76 | 0.05 |

### *4.2. Model Training*

All models were trained by utilizing a designated training set. At the end of each training epoch, the DICE performance on the independent validation set was computed.

Throughout the training process, the model with the best validation DICE performance was saved. The best model was then used to compute the DICE and bAVD metrics on the test dataset to learn the true model performance.

Figure 6 shows the behavior of the training loss, validation loss, and validation DICE throughout the training process. The curves were generated using TensorBoard [37]. For all curves, the x-axis displays the training steps, and the y-axis shows the value of the loss or, in the case of the validation DICE, the average DICE coefficient score. Due to the differences in batch sizes and model parameters, the speed of convergence and the number of training steps varied. The training and validation loss curves used the logarithmic scale on the y-axis to show the change in more detail as the model converged. The graphs indicate that training and validation loss trended to convergence for each model. In addition, as the models converged, we observed the improving trajectory of the DICE score in validation set prediction. The figure displays that the GWC graphs were much more irregular than the others. The spikes are typical for the Adam optimizer and mini-batch optimization. Additionally, the training obtained using Atropos consisted of many inaccuracies, which hurt the convergence process.

### 4.3. Pipeline Performance

### 4.3.1. Runtime

PigSNIPE is designed for running single images. The user can specify the pipeline configuration to choose what analysis to perform. Additionally, the user can specify if the pipeline will execute using the GPU or CPU only. GPU usage can speed up this process by around 10–15 s. However, the current design of the pipeline is not aimed at fully leveraging GPU execution.

Table 7 shows the runtime of PigSNIPE. The results were obtained by running PigSNIPE on the subset of the Iowa and Germany datasets. The pipeline was in its full configuration, executed inside a docker container and using the CPU to simulate the tool's expected most common usage. The runtime was closely related to the file sizes. The German images were much bigger than the Iowa images, with the T2w images being twice the size and T1w images being up to six times the size. The large file sizes negatively impacted the pipeline performance, resulting in an execution time for the German data of 5 min compared with 2 min for the Iowa data. We found that most of the runtime was spent on registration and landmark detection in all cases. The registration took around 30 s for the Iowa data and up to 3 min for the German data. The landmark detection took around a minute for the Iowa data and up to 2 min for the German data.

**Table 7.** Pipeline's runtime analysis. Running PigSNIPE using CPU inside a Docker container took around two minutes for Iowa and 5 min for German datasets to perform full analysis. Image files in the German dataset were much larger than those in the Iowa dataset, leading to an increased runtime.

| | T1w Size (MB) | | T2w Size (MB) | | Runtime (s) | |
|---|---|---|---|---|---|---|
| Dataset | Mean | SD | Mean | SD | Mean | SD |
| Iowa | 23.37 | 0.28 | 22.34 | 0.37 | **129.01** | 6.14 |
| Germany | 129.67 | 25.52 | 46.27 | 5.23 | **293.32** | 10.98 |

### 4.3.2. Memory and Hardware

The overall memory requirement to support full pipeline capabilities was 11 GB. The majority of that was for the reinforcement learning models, taking up 9 GB. The GPUs used for training the models were NVIDIA GTX8000 and NVIDIA GV100, and the running of the pipeline was executed on an Intel Xeon with 56 cores and 72 threads. However, all processes, except registration, were executed using a single core.

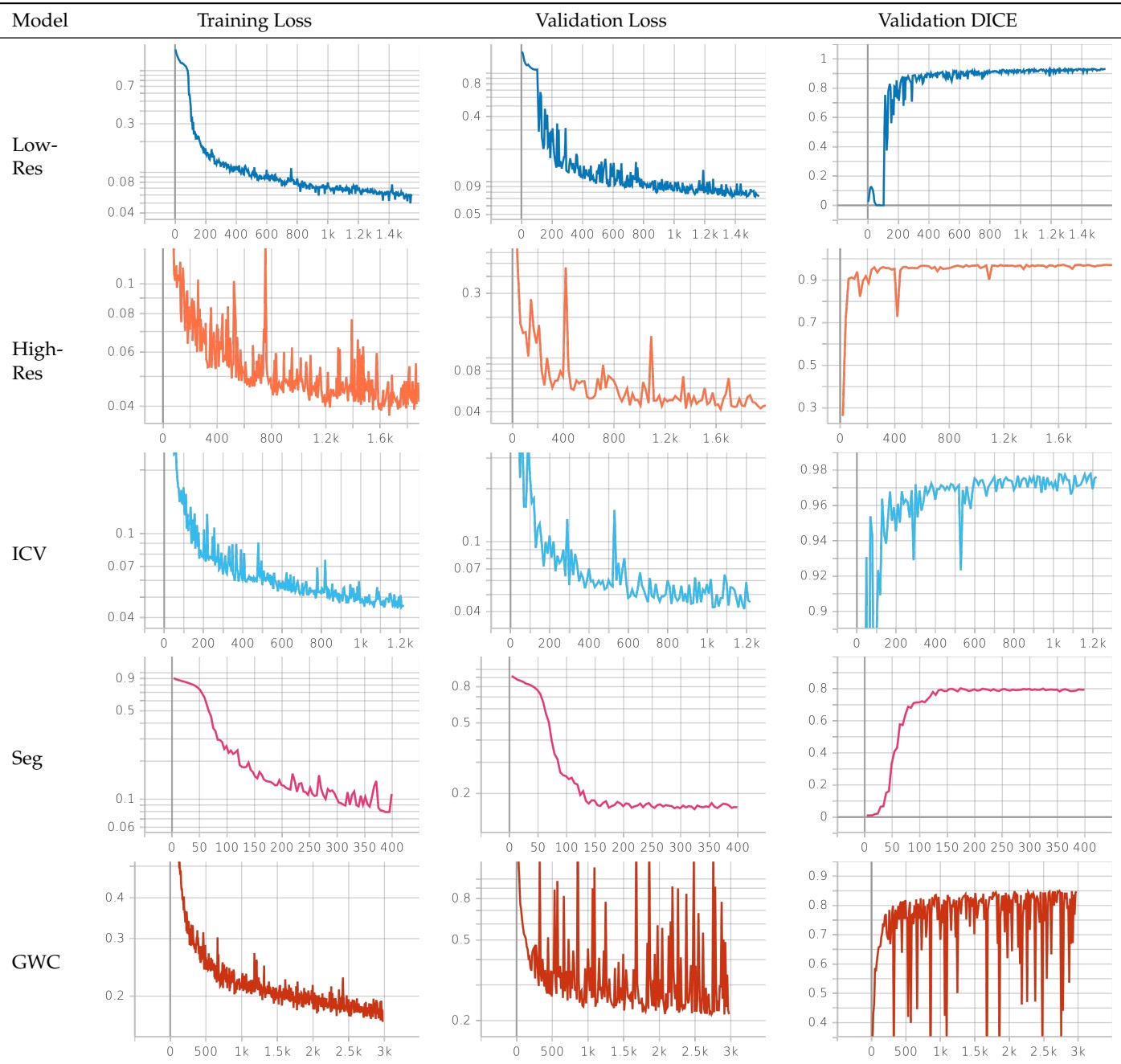

**Figure 6.** Training convergence curves. Training and validation losses are shown with a logarithmic scale and indicate that all models converged properly. The validation DICE curves use a linear scale and display improvement with the model convergence score.

## 5. Discussion

We present MinipigBrain, a deep learning-based pipeline for minipig MRI processing. Trained on heterogeneous datasets, the pipeline can perform multiple operations commonly used in human neurodegenerative research for application in minipig models.

### 5.1. Deep Learning Models

Splitting the initial brainmask process into low- and high-resolution masks was necessary to extract a region of interest from the original image space. The low-resolution masks proved robust and sufficiently accurate for preparing the data for the subsequent processes.

The caudate-putamen segmentation model accuracy was worse than those of the high-resolution and ICV models, which produced highly precise masks. The caudate and putamen are small, hard-to-distinguish regions. Even minor errors in segmenting of such small regions could lead to a higher impact on the DICE metric. Additionally, we predicted that the worse quality of the T2w images in the Germany dataset harmed the training process, and we are investigating ways to improve the model's performance.

The evaluation of the tissue classification model was more complicated. We did not have the resources to obtain high-quality manual labels for the gray and white matter and cerebrospinal fluid segmentation problem and therefore used the Atropos results as the "ground truth". However, Atropos is a tool aimed at human data rather than minipigs, and it produced suboptimal results. During visual qualitative analysis, we found examples of mistakes in the Atropos images. This negatively impacted the evaluation of our model, as the DICE metric only measures the overlap between the masks and not the quality of the masks. However, despite the suboptimal labels, during the training process, the model was able to generalize and produce arguably better results than the "ground truth" masks in many cases. During the visual qualitative review, a trained professional preferred our model's prediction over that of Atropos. We acknowledge the shortcomings of our algorithm in the tissue segmentation problem, and we are actively exploring ways to improve the model's performance. Nonetheless, we believe that our model still presents scientifically valuable results that can be useful to researchers.

### 5.2. Pipeline Performance

To the best of our knowledge, PigSNIPE is the first medical imaging tool aimed at large animals that allows for many of the most common processes. Its current form can deliver consistent results in under five minutes per scanning session. Currently, researchers are forced to analyze datasets manually. Performing all the operations available in PigSNIPE can easily take a trained professional a full day of work per pair of T1w and T2w images. Therefore, our tool can drastically decrease the time to perform minipig image analysis, significantly reducing research costs and accelerating scientific discovery.

### 5.3. Limitations

PigSNIPE is aimed at neuroscientists using large animal subjects in their research. As clinical interventions for animals are minimal, our tool has no immediate clinical utility.

We designed PigSNIPE for ease of use by providing the set-up steps and execution through a Docker container, which further simplified the configuration on the user side. However, using PigSNIPE requires a Linux environment, Docker, and basic skills with the bash shell.

The pipeline is constructed to apply to similarly shaped large quadrupeds (sheep, goats, etc.), but lack of access to those data prevented validation across species.

### 5.4. Future Work

To improve the performance of segmentation and tissue classification, we plan to experiment with transfer learning. By leveraging a large human dataset and nonlinear transformation, we expect to improve the models' accuracy.

Additionally, we plan to add a batch execution capability to the pipeline. Loading a model onto the GPU takes more time than the inference time. We have nine different deep learning models, and being able to execute images from different subjects in a batch should significantly speed up the execution time.

**Author Contributions:** Conceptualization, M.B. and H.J.J.; methodology, M.B.; software, M.B.; validation, M.B. and H.J.J.; formal analysis, M.B.; investigation, M.B.; resources, H.J.J. and J.C.S.; data curation, M.B. and K.K.; writing—original draft preparation, M.B.; writing—review and editing, M.B. and H.J.J.; visualization, M.B.; supervision, H.J.J.; project administration, H.J.J. and J.C.S.; funding acquisition, H.J.J. and J.C.S. All authors have read and agreed to the published version of the manuscript.

**Funding:** Funding for data collection of the Iowa datasets was provided by the Noah's Hope and Hope 4 Bridget Foundations. Imaging data collection was conducted with an MRI instrument funded by NIH 1S10OD025025-01. Funding for collection of the German imaging data was provided by the CHDI Foundation and partners who support the work of the George-Huntington Institute and thus provided additional funding for the initial collection of these datasets.

**Institutional Review Board Statement:** Germany Data: All research and animal care procedures used to collect these data were approved by the Landesamt für Natur und Umweltschutz, Nordrhein-Westfalen, Germany [84-02.04.2011.A160] [14]. Iowa Data: All procedures used to collect these data were approved by the Institutional Animal Care and Use Committees (IACUC) of the University of Iowa and Exemplar Genetics in accordance with regulations [15].

**Data Availability Statement:** Not applicable.

**Acknowledgments:** For their work on the German dataset, we want to acknowledge Ralf Reilmann, who secured funding and oversaw the methodological development of MR imaging protocols, and Robin Schubert, who worked on data acquisition and processing. In addition, we would like to thank Jill Weimer, the principal investigator of Noah's Hope, for her help in securing funding for the Iowa dataset collection. Finally, we would also like to thank Alexander B. Powers for his help with the reinforcement learning implementation for landmark detection and Jerzy Twarowski and Ariel Wooden for their support and figure design critiques.

**Conflicts of Interest:** The authors declare no conflict of interest.

## Abbreviations

The following abbreviations are used in this manuscript:

| | |
|---|---|
| T1w | T1-weighted MRI sequence |
| T2w | T2-weighted MRI sequence |
| CSF | Cerebrospinal fluid |
| GWC | Gray matter-white matter-CSF |
| Seg | Caudate-putamen segmentation |
| bAVD | Balanced average Hausdorff distance |

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
