# Peer review of "PigSNIPE: Scalable Neuroimaging Processing Engine for Minipig MRI"

_algorithms, doi:10.3390/a16020116_

Round 1

Reviewer 1 Report

I have reviewed the manuscript the overall contents of this manuscript is well organized to give a clear overview of this work. I have suggested some comments about this work are as the following:

Comments to the Authors:

1.     Authors should write clearly abstract including background, method, results with numerical values and conclusion.

2.     Authors should revise the introduction section in three paragraph, first paragraph for Neurological disorders, second paragraph for MRI, third paragraph for deep learning model and last paragraph for objective of this study and hypothesis.   

3.     In method section does author used any statistical analysis method ?.

4.     In results, section author should write the figure captions/legends in more details from Figure 1 to Figure 5. Explain clearly about the main findings of the figures.

5.     Author should revise the discussion of this study, like how and why this algorithm is better than previous deep learning studies. It has written in very short form. 

6.     The authors should write some limitations of this study and clinical application in more details.

Author Response

Comments and Suggestions for Authors 

I have reviewed the manuscript the overall contents of this manuscript is well organized to give a clear overview of this work. I have suggested some comments about this work are as the following: 

We sincerely thank you for your review and suggested improvements. We agree with your critique and have implemented changes as you suggested. We would like to thank you for your help in making this manuscript better. Below are responses to your suggestions and description of changes we made accordingly. 

Comments to the Authors: 

1.     Authors should write clearly abstract including background, method, results with numerical values and conclusion. 

Changes in Abstract. 

We rewrote the abstract to include information about background, methods, and results and organized it in a clear fashion. 

2.     Authors should revise the introduction section in three paragraph, first paragraph for Neurological disorders, second paragraph for MRI, third paragraph for deep learning model and last paragraph for objective of this study and hypothesis.    

Changes in Introduction. 

We rewrote the introduction. It is now in 4 paragraphs for neurological disorders, second for MRI and medical imaging, third for deep learning, and last for our model and study objective. 

3.     In method section does author used any statistical analysis method? 

Changes in Results. 

We modified our results section to add information on the standard deviation of computed metrics to better show the consistency of our models performance.  

Additionally, we do not use statistical analysis in our methods as our model is not statistically driven.  

4.     In results, section author should write the figure captions/legends in more details from Figure 1 to Figure 5 Explain clearly about the main findings of the figures. 

Changes throughout the document. 

We fully agree with this point. We rewrote all captions for figures and tables to provide the presented data's main findings clearly.  

  

5.     Author should revise the discussion of this study, like how and why this algorithm is better than previous deep learning studies. It has written in very short form.  

Changes in sections 4.1.1 and 4.1.4 

Currently there is no other tools aimed at minipig images that we could compare to. However we used SynthStrip - the state-of-the-art algorithm for human data - and applied it to the minipig data. The algorithm failed entirely, demonstrating that human data-oriented software is unsuitable for animal data.  

Additionally, we rewrote the Gray-White-CSF Segmentation Mask part. We think it was misleading to describe Atropos' results as “ground truth”. Therefore, we improved that section of the paper and extended the discussion section for this part. 

  

6.     The authors should write some limitations of this study and clinical application in more details. 

We have added section 5.3 - Limitations to the Discussion part of the paper where we included that information and other limitations of our algorithm. 

Reviewer 2 Report

This paper proposed a PigSNIPE pipeline to automatically handle, process, and analyze minipig MR images from two real datasets. They aimed to register medical images, align AC-PC, detect landmarks, strip skulls, and lesion area segmentation for monitoring animal disease development. Experimental results on the Germanly and Iowa datasets demonstrate the good performance of the proposed pipeline.  Overall, this paper is well organized, here are some specific comments for minor revision.

Q1. The pipeline in fig.1 is so complicated that it is not easy to understand how the proposed pipeline works since there are so many arrows and abbreviations.  I suggest the authors simplify the pipeline's dataflow into several main modules and detail these modules in other figures. Besides, it's better to detail the network structure of deep learning models.

Q2. In the result section, the details of the model training are not clear, as the convergence of the proposed model is important to the prediction performance. The authors should add the training and testing loss curve in this paper. 

Q3.  The segmentation results in fig.3 and fig.4 need to be improved. The qualitative segmentation result is not clear compared with the ground truth. It is better to increase the contrast of the segmented area. Also, other related models should be added to show the effectiveness of this work.

Q4.  MR image computing, including detection, segmentation etc., using deep learning is a hot topic and authors should give a more extensive review in this field. There are some representative works, such as:

Fine Perceptive GANs for Brain MR Image Super-Resolution in Wavelet Domain;

Automatic recognition of mild cognitive impairment from MRI images using expedited convolutional neural networks;

Cross-modality Synthesis from MRI to PET Using Adversarial U-Net with Different Normalization

Bidirectional Mapping Generative Adversarial Networks for Brain MR to PET Synthesis.

Q5. The authors claimed that the proposed pipeline has obvious advantages in computational efficiency. But they didn't give quantitative value and error of calculation time. Could the authors give more details on the computational efficiency in this paper?

Q6. The paper is well organized generally. However, there are some grammatical errors in different parts of the text. For example, in line 53, the word "mm" in the sentence "... the acquisition resolution of 0.5x0.514x0.514 mm ..." should be "mm3";  The word "manually" in line 238 is repeated. The formatting of the paper needs to be checked carefully.

Author Response

This paper proposed a PigSNIPE pipeline to automatically handle, process, and analyze minipig MR images from two real datasets. They aimed to register medical images, align AC-PC, detect landmarks, strip skulls, and lesion area segmentation for monitoring animal disease development. Experimental results on the Germanly and Iowa datasets demonstrate the good performance of the proposed pipeline.  Overall, this paper is well organized, here are some specific comments for minor revision. 

We sincerely thank you for your review and suggested improvements. We agree with your critique and e would like to thank you for your help in making this manuscript better. Below are responses to your suggestions and description of changes we made accordingly. 

Q1. The pipeline in fig.1 is so complicated that it is not easy to understand how the proposed pipeline works since there are so many arrows and abbreviations.  I suggest the authors simplify the pipeline's dataflow into several main modules and detail these modules in other figures. Besides, it's better to detail the network structure of deep learning models. 

Changes implemented in sections 3.1, 3.2 and Fig 1. 

In meeting all reviewers desires we decided not to change figure 1. However, to mitigate this issue, we rewrote the caption of the figure and the section describing the pipeline architecture to explain the figure and make understanding the dataflow easier.  

Additionally, insection 3.2, we clarified where the detailed information on the architecture can be found. 

Q2. In the result section, the details of the model training are not clear, as the convergence of the proposed model is important to the prediction performance. The authors should add the training and testing loss curve in this paper.  

Added section 4.2 - Model Training. 

We added the “Model Training” part in the results section. There we plot the convergence curve for training and validation loss and the validation dice score to show how the models converge. 

Q3.  The segmentation results in fig.3 and fig.4 need to be improved. The qualitative segmentation result is not clear compared with the ground truth. It is better to increase the contrast of the segmented area. Also, other related models should be added to show the effectiveness of this work. 

Changes in figure 3 and 4 (and corresponding sections) and in section 5.1 paragraph 3. 

Fig 3 now better shows the underlying structure of the Caudate and Putamen and better displays the example difference between ground truth and model prediction.  

In Figure 4 we decided to showcase Axial instead of Sagittal slices as we believe it better displays both model performance. Additionally, we rewrote the Gray-White-CSF Segmentation Mask part. We think it was misleading to describe Atropos' results as “ground truth”. Therefore, we improved that section of the paper and extended the discussion section for this part.  

Q4.  MR image computing, including detection, segmentation etc., using deep learning is a hot topic and authors should give a more extensive review in this field. There are some representative works, such as: 

  • Fine Perceptive GANs for Brain MR Image Super-Resolution in Wavelet Domain; 

  • Automatic recognition of mild cognitive impairment from MRI images using expedited convolutional neural networks; 

  • Cross-modality Synthesis from MRI to PET Using Adversarial U-Net with Different Normalization 

  • Bidirectional Mapping Generative Adversarial Networks for Brain MR to PET Synthesis. 

Changes in Introduction, paragraph 3. 

We support this critique. We rewrote the introduction to include a paragraph on the Deep Learning in medical imaging. We also thank you for recommending some of the fine articles in our field. After familiarizing ourselves with this work we cited an article you recommended. 

Q5. The authors claimed that the proposed pipeline has obvious advantages in computational efficiency. But they didn't give quantitative value and error of calculation time. Could the authors give more details on the computational efficiency in this paper? 

Changes in sections 4.3.1 and 5.2 

We extensively modified the Pipeline Performance part of the results. We included a table with means and the standard deviation of the image file sizes and the pipeline’s runtime. Additionally, we compare our model to the manual processing time as it is currently the only other available option for minipig image processing. 

Q6. The paper is well organized generally. However, there are some grammatical errors in different parts of the text. For example, in line 53, the word "mm" in the sentence "... the acquisition resolution of 0.5x0.514x0.514 mm ..." should be "mm3";  The word "manually" in line 238 is repeated. The formatting of the paper needs to be checked carefully. 

Changes throughout the document. 

We agree with this critique. To mitigate this issue and improve the quality of writing, we put the whole text through Grammarly Pro and made correction as detected by the tool. 

Reviewer 3 Report

This is a well designed, well performed and well written paper on a niche topic that is nevertheless clinically relevant. The topic of automatic handling and processing of MRI data of the brain in animal experiments, here minipigs, is important for preclinical research. The authors apply their tool, called PigSNIPE, which was developed and described earlier, to two different test data sets and report good results. 

Author Response

We sincerely thank you for your review. We appreciate your time and expertise.